# From Muscle to the Myofascial Unit: Current Evidence and Future Perspectives

**DOI:** 10.3390/ijms24054527

**Published:** 2023-02-24

**Authors:** Antonio Stecco, Federico Giordani, Caterina Fede, Carmelo Pirri, Raffaele De Caro, Carla Stecco

**Affiliations:** 1Department of Rehabilitation Medicine, New York University Grossman School of Medicine, New York, NY 10016, USA; 2Department of Rehabilitation Medicine, Padova University, 35141 Padova, Italy; 3Department of Neuroscience, Institute of Human Anatomy, University of Padova, 35141 Padova, Italy

**Keywords:** myofascial unit, motor unit, fascia, motor control, review, muscle, muscle spindle, connective tissue

## Abstract

The “motor unit” or the “muscle” has long been considered the quantal element in the control of movement. However, in recent years new research has proved the strong interaction between muscle fibers and intramuscular connective tissue, and between muscles and fasciae, suggesting that the muscles can no longer be considered the only elements that organize movement. In addition, innervation and vascularization of muscle is strongly connected with intramuscular connective tissue. This awareness induced Luigi Stecco, in 2002, to create a new term, the “myofascial unit”, to describe the bilateral dependent relationship, both anatomical and functional, that occurs between fascia, muscle and accessory elements. The aim of this narrative review is to understand the scientific support for this new term, and whether it is actually correct to consider the myofascial unit the physiological basic element for peripheral motor control.

## 1. Introduction

The “motor unit” has long been considered the quantal element in the control of movement formed by both the motoneuron and the muscle fibers that its axon innervates. However, a single motor unit never works alone; it is always in synergy, not only with other motor units of the same muscle, but also with different synergistic muscles [1]. Furthermore, in the last few decades, the study of the fascia has led to questioning of several paradigms of human anatomy and physiology. In particular, the role of fasciae working as a bridge between various muscles involved in the same movement was demonstrated by many authors [2,3], inducing Luigi Stecco, in 2002, to create a new term, the “myofascial unit” (MFU), to describe the bilateral dependent relationship, both anatomical and functional, that occurs between fascia, muscle and accessory elements, such as vessels, joint proprioceptors, ligaments and tendons [4]. More precisely, he described the myofascial unit as an entity that includes the motor units involved in a specific movement, the associated connective tissue, the nerves that give muscular contraction input and movement perception, and blood and lymphatic vessels supplying the area involved. For the author, this unit is the basis of peripheral motor coordination and dynamic proprioception. This term was also used for the first time in 2022 by the National Center for Complementary and Integrative Health (NCCIH) to announce a grant for further investigation of the “Myofascial unit”, but without giving a clear definition of what a myofascial unit is.

This narrative review aims to better define the myofascial unit in all its components (muscle, connective tissue, nerves, vessels) from a histological, molecular, neurophysiological, and biomechanical point of view, to better understand the anatomy and physiology of this entity.

## 2. Muscle

The classical point of view is based mainly on a segmental approach where patterns of movement are modeled in a linear framework of isolated muscle groups which provide joint action according to the muscle insertions. Trotter and Purslow [5], in 1992, demonstrated that muscle fibers are totally embedded in the intramuscular connective tissue, and endomysium, perimysium, epimysium and deep fascia generate a continuum from microscopic to macroscopic patterns. From an anatomical point of view, the innermost layer (endomysium) is directly in contact with the sarcolemma, and therefore, with every single muscle fiber [6]. Hence, every force generated by the muscular fiber is transmitted directly to the endomysium thanks to this internal connective structure composed of proteins (such as dystrophin and integrin) that cross the sarcolemma [6,7]. Together with the sarcolemma, the endomysium resists longitudinal deformation of the myofiber only when it has exceeded 150% of its physiological length, thus outside normal movement conditions. Conversely, the endomysium is resistant if the forces of traction have a transversal pattern. This observation explains why length changes in non-activated muscle fibers follow the length changes of actively contracting neighboring fibers when only a subset of muscle motor units is activated in sub-maximal contraction [6]. The muscle fibers that are not recruited become, thanks to the endomysium, a real tendon for the transmission of lateral force without having to change the length [8].

The endomysium extends without interruption in the perimysium’s collagen. The latter divides the muscle belly into fascicles of different dimensions (primary or secondary fascicles) and forms a continuous network across the width of a muscle and from the origin to the insertion of fascicles. Inside the perimysium structure, immersed in a matrix of proteoglycans, there are collagen fibers type I, III, IV, V, VI, and XII [9,10]. Collagen type I (COLI)provides the perimysium with a notable resistance to traction. It is this part of the intramuscular connective tissue that has a fundamental role in transmission of the force generated in the muscle toward the bone levers [11]. Thus, the perimysium structure does not contribute to the passive rigidity of the muscle but is rather an organized framework to transmit the forces produced in the locomotor system [12]. It also allows connection between synergic muscle fibers of different fascicles, an attachment for the muscle fibers that do not end in the tendon and it guarantees independence between muscle fascicles during muscle contraction [13]. Fede et al., (2022) [14] analyzed the effect of aging on the intramuscular connective tissue of humans and mice, demonstrating that with aging there is a huge increase in the percentage area of collagen (mainly due to an increase in COLI), and a decrease in the percentage area of elastic fibers in the perimysium, together with a significant decrease in hyaluronan content. No significant differences were detected according to gender. These changes could cause a stiffening of the muscles and a reduction of their adaptability that are typical symptoms of aged people. These results also fit with the study by Pavan et al., (2020) [15], that compared the passive stress generated by elongation of fibers alone with that of fibers arranged in small bundles in young and healthy versus elderly subjects. The authors demonstrated that the resistance to elongation of fibers alone was not different between the two groups, while the reduced compliance was due to the contribution of ECM (extracellular matrix) in elderly samples compared to young ones. This experiment clearly supports the muscle fibers and the intramuscular connective tissue forming a unit, and the alteration of one element easily also affecting the other one. Therefore, we cannot speak about muscular aging or muscular stiffness without also taking into consideration the surrounding intramuscular connective tissue.

As the perimysium approaches the surface of the muscle, it merges seamlessly with the epimysium [11]. The epimysium is thicker than the other elements of the intramuscular tissue and is formed by collagen fibers with a larger diameter [16]. It covers all the muscle bellies, forming a lamina that clearly defines the volume of each muscle, but it also merges into the paratenon of the tendons, creating further continuity between muscle and tendon [17]. In fusiform muscles, the collagen of the epimysium resists passive elongation to the limits of the physiological deformation of the muscle. In pennate muscles, the collagen fibers mainly reflect the progression of muscular fibers, forming a dense lamina that often acts as a superficial tendon or an aponeurosis that inserts itself into the connective tissue of the locomotor system or in the adjacent muscles [11]. The epimysium provides a clear resistance if the tension is in the same direction as the fiber’s trend. However, it is possible to observe a discrete yielding if the traction occurs in an orthogonal manner [8]. This is due to the distribution of the collagen fibers of the fascial system that are not arranged only parallel to the muscle fibers, but also in other directions [18].

The deep fascia is connected to the underlying muscles thanks to many myofascial expansions, and it represents a parallel pathway for force transmission beyond the muscle, tendon, and bone [19,20,21,22] through which more than 30% of the mechanical forces are transmitted. These myofascial connections create a continuity among the muscles involved in the same movement, representing the anatomical base of the myofascial chain. The timing and intensity of each muscle segment’s activation are coordinated across muscles and influenced by the muscle segment’s moment arm and its mechanical line of action in relation to the intended movement direction [23]. It has been shown that multiple muscle fibers in neighboring muscles are synchrony-activated during specific motion, and traction of one single muscle bundle can stretch or pull other muscle bundles localized in adjacent muscles thanks to the intramuscular myofascial force transmission [23]. The latter occurs through the surrounding connective tissue. Furthermore, only 70% of the muscle fibers have a tendinous insertion, while 30% have a fascial insertion [24], which allows muscle tension to be transmitted onto the fascia [25]. Thus, there are several limits in the classical muscle subdivision that do not reflect biomechanics when related to a specific movement. In turn, the direct morphological and functional continuum between muscles and fasciae is at the base of the mechanical interactions between agonist muscles as well as between antagonistic muscles [26]. If the connections between the motor unit and the connective tissue are not considered, an important component of the system is excluded. In a recent study [27], the MRI (magnetic resonance imaging) evaluation of 11 competitive athletes with adductor magnus injury demonstrated that four of them involved the myofascial junction, and not the tendon or the muscle. Cruz-Montecinos et al., (2022) [28] demonstrated a significant interaction among the soleus muscle stiffness and knee position, as if soleus, according to the classical biomechanical models, is a monoarticular muscle. Again, only if we consider all the interactions due to the fascial structure, can we easily explain these clinical findings and appropriately manage patients.

In addition to direct intermuscular connections generated by collagenous linkages between epimysium of adjacent muscles [3,29], indirect intermuscular connections (e.g., neurovascular tracts) are also present. It is important to underline that the compartmental boundaries (e.g., intermuscular septa, interosseal membranes, periosteum, and compartmental fascia) are continuous with the neurovascular tracts. These extra-muscular connections bind muscular and non-muscular tissues at several locations and are the anatomical substrate for the epimuscular force transmission that has already been demonstrated to occur between both synergic and antagonist muscles [20,30,31].

## 3. Nerve and Muscle Spindle

The classical definition of motor unit is: “A motor unit is made up of a motor neuron and all of the skeletal muscle fibers innervated by the neuron’s axon terminals, including the neuromuscular junctions between the neuron and the fibres” [32]. In this definition, the role of the intramuscular connective tissue is not considered; however, it is well known that to analyze the morphology of the neuromuscular junction it is important to remove all the intramuscular connective tissue that envelopes the muscular fiber and the neuromuscular plate [33]. In addition, to plan and correctly execute movements, we need both a motor input and good motor control based on proprioceptive input. This principle applies to low levels of the hierarchy, such as spinal reflexes, and to higher levels. Several types of sensory receptors lie within skeletal muscles, including the muscle spindles and Golgi tendon organs that monitor muscle contraction, as well as the free nerve endings that are slow conducting, thinly myelinated, or unmyelinated nerve afferents sensitive to mechanical, chemical, and nociceptive stimuli. All these elements were considered to be related to the muscle, but recently many papers have highlighted that they have strong connections with the intramuscular connective tissue and fasciae.

The muscle spindle (MS) is a key element for the regulation of muscle contraction. Traditionally seen as a feedback regulator system (stretch-sensitive organ) [34], the MS has recently gained importance as a feedforward element [35]. They are traditionally considered as part of the muscles, but they are totally embedded in the intramuscular connective tissue and their sensitivity is strongly influenced by it [36,37]. Indeed, the intrafusal fibers of the MS possess intracapsular terminations or extend beyond the limit of the MS capsule terminating in the intramuscular connective tissue of adjacent extrafusal fibers [38]. In addition, other studies have demonstrated the continuity of the outer capsule with the ECM of extrafusal fibers and with the perineural epithelium [37,39,40].

The intrafusal fibers provide tension to the spindle in order to maintain tautness and are therefore sensitive to stretch across a wide range of motion. When a muscle contracts, two motor nerves from the central nervous system are involved: the alpha motor neuron, which is responsible for the muscle contraction, and the gamma motor neuron, whose main responsibility is to activate the MS. The activation of the intrafusal fibers will stretch not only the capsule of the MS, but also the adjoining perimysium and epimysium. The contrary is also possible: the deformation of the intramuscular connective tissue could be transmitted to the spindle capsule, stimulating the annulospiral endings of the Ia fibers and type II fibers, all of which generate input to the spinal cord. If no peripheral or central inhibition occurs, activation of the alpha motor neuron will take place. This activation will generate the contraction of the extrafusal fiber of the motor units, creating what is called a gamma loop, essentially a feedback loop that regulates muscle tension. Through the connection between the capsule of MS and the connective tissue, the function of the MS can be affected by a tensioning of the perimysium: abnormal tension can create dysfunction in the velocity, modality, and ability of capsule shortening. These factors are in direct relation to the quality of both the intramuscular and surrounding connective tissue such as intermuscular septa and deep fascia. This phenomenon guarantees the possibility to analyze the adaptability of the surrounding environment which is then translated into a different stimulation of the annulospiral and flower spray endings. We consider this event fundamental to modulate the activation of the corresponding motor unit, leading to motor control dysfunctions. Beyond the normal neurodegeneration and morphological alterations of the MSs, age-related physiological changes could also be explained by alterations in the environment directly surrounding them. A recent work by Fan and coauthors (2022) demonstrated that older adult mice have thicker MS capsules, with an accumulation of COLI type and a decrease in hyaluronan [37]. These age-related changes may reduce the ability to adapt to stretching and to feel the increase of tension of the associated muscular fibers, resulting in a higher threshold and in failed motor unit recruitment. Consequently, a reduced motor unit activity may lead to a lower contractile strength of the muscle, compromising the regulation of muscle tone. The important role of the connective tissue again highlights the idea of considering these structures as a unit working together. James et al., (2022) [36] confirmed this hypothesis in an animal model. They analyzed the MSs in the multifidus muscle of healthy sheep and in sheep with intervertebral disc (IVD) degeneration. They did not found differences in the number or location of MSs after IVD degeneration, but there was a thickening of connective tissue surrounding the MS due to increased expression of Collagen I and III. The authors suggested that the changes in the connective tissue and collagen expression of the MS capsule are likely to impact their mechanical properties, and consequently the transmission of length change to MSs, and thus transduction of sensory information. This change in MS structure may explain some of the proprioceptive deficits identified in low back pain. Furthermore, the intrafusal muscle fiber contraction spreads its load to the surrounding intramuscular connective tissue, reaching not only the surface of its muscle but also the surface of neighbor synergic muscles. Even if minimal, this load could influence the capsule tensile state of other MSs and modulate their afferent signal to the spinal cord (through IA fiber) where synapse with the alfa motoneuron occurs. At this point, the efferent signal will reach the motor plate of the different motor units to generate the final muscle contraction. This sophisticated infrastructure presents many advantages from injury prevention to modulation of the activation of muscle fibers according to gesture performance. The ability to investigate the adaptability of the muscle that is performing a massive contraction will prevent possible tears at the muscle level. Furthermore, it allows muscle contraction to adapt to the constant variation that environment and joint leverage require, such as high-performance movements that are not simple antigravitation but require a constant remodulation such as in sports. On the other hand, when perimysium properties are altered, as demonstrated after limb immobilization [41], MSs connected to the perimysium will not function correctly. If the perimysium is rigid, the MSs and the tendon Golgi organs are unable to change their length, and consequently, they cannot be activated [42]. Changes in the perimysium quality as a result of trauma, poor posture, post-surgery, or overuse, may cause the inhibition of normal spindle cell stretching and result in abnormal feedback to the central nervous system leading to impaired regulation of the muscle tone. In addition, the changes in the fascia system in immobilized skeletal muscle seem to contribute to alterations in the biomechanical properties of the entire muscle belly, reducing its compliance [41]. This, in turn, increases the stretch reflex of the MS as the pull is transmitted more efficiently to the MSs in a less extensible muscle [43]. Siegfried Mense, one of the world’s leading experts on muscle pain and neurophysiology, was questioned about the possible role of connective tissue alterations on muscle spindles, and answered “Structural disorders of the fascia can surely distort the information sent by the spindles to the CNS and thus can interfere with a proper coordinated movement” and “the primary spindle afferents (Ia fibers) are so sensitive that even slight distortions of the perimysium will change their discharge frequency”.

To have a good proprioception and precise feedback of movement, and also to perceive pressure pain and other pain input, the free nerve endings of the muscle are also important. They were extensively described by Stacey in 1969 [44]. He analyzed various muscles of cats, highlighting inside the muscle myelinated nerve fibers small in diameter (lower group II and III), and he demonstrated that two-thirds of the sensory component in a muscle consists of non-myelinated nerve fibers with a diameter range between 0.5 and 1.25 microns. They regularly branch, creating smaller nerve fibers that end freely in the muscle and correspond to the C afferents referred to in the Literature. Actually, if we look at the anatomical description of these free nerve endings inside the muscle, the author affirms that “the nerves form an anastomosing plexus in the muscle connective tissue”, suggesting as, when we are speaking about free nerve endings in the muscles, they are actually located in the intramuscular connective tissue, and that probably the conditions of the ICT can affect the sensitivity of this plexus. These findings were confirmed recently by Fede et al. [45,46] who demonstrated a nerve network inside the thoracolumbar and the gluteal fasciae (Figure 1). The density of these two networks is totally different, having much more sensitive fibers inside the thoracolumbar fascia, whilst the autonomic nerve fibers are equally present in the two fasciae. In addition, the authors affirmed that the deep fascia is much more innerved with respect to the muscle. Three further studies have compared the density of innervation of the deep fascia and of the underlying muscles, demonstrating in all cases that the fasciae result in significantly more densely innervated TLF with respect to latissimus dorsi [47], masseter fascia and muscle [48], and quadriceps muscle and fascia lata [45]. In Stacey’s work, we can also find similar results. Indeed, he underlines that the density of the plexus of nerve bundles is dense in the skin, less dense in the periosteum, and least in the fascia. Finally, the peri-vascular nerve trunks in the muscle form an extremely loose plexus. This evidence is in line with the suggestion that the fascia can be a sensory organ and that, on many occasions, myofascial pain could be caused by a fascial alteration, rather than a muscular problem.

Many studies demonstrate the importance of mechanoreceptors in the fascial layers for their proprioceptive capacity [49], with a particularly dense innervation of the superficial layers of the deep fascia [50,51]. The key nerve elements are the free nerve endings (they may be up to seven times more numerous than the other mechanoreceptors), but in the plantar and palmar fascia and in some retinacula also, Ruffini and Pacinian corpuscles are present [52,53]. All of these structures are totally embedded in the fascial connective tissue and feel any tensional variation in the fascial structure or any stretch to the muscle and fascia (Figure 1).

All of these elements further the importance of fascia as a sensory system [54]. Moreover, because of its dense sensory innervation, including nociceptive fibers, the fascia can play an important role in pain perception, and the pain from the fascia can be even more aggravating than pain from the muscles. Schilder and co-authors reported that fascial pain is usually described as a burning, cutting, or stinging sensation, whereas muscle pain is described as a duller pounding and beating [55,56]. The higher pain-sensitivity and pain areas support fascia as a fundamental contributor to pain mechanisms and perception. Some recent research demonstrated in rats how an inflamed thoracolumbar fascia shows an increase in nociceptive fibers compared to an intact fascia, highlighting the pathological altered fascia as a source of pain [57].

In addition, it is important to note that the perineurium of a nerve is totally in continuity with the epimysium of the surrounding muscles, as demonstrated by Stecco et al., (2020) [58]. In this way, an unbalanced tension of the epimysial fascia can affect the paraneural sheath, also limiting the nerve displacement.

## 4. Blood Vessels

The vascular inflow to skeletal muscles is provided by feed arteries that are usually distributed in the epimysium along the long axis of the muscle. They account for as much as 30–50% of the total resistance to blood flow through skeletal muscle, representing a significant site for blood flow control [59]. Secondary arteriolar branches divide at right angles to these feed vessels and enter the perimysium and travel perpendicular to the muscle fiber axis until giving rise to terminal branches that branch into numerous capillaries that are embedded in the endomysium and travel parallel to the muscle fiber [60]. From this description it is evident that all the vessels for muscles are completely embedded in the epimysium, perimysium, and endomysium. In addition, Pirri et al., (2022) [61] described the rich vascular pattern of the fasciae, which forms fine, dense meshwork with an area of 6.20% ± 2.10%. The diameters of the vessels fall between 13 and 65 μm. The network is composed of arteries, veins, capillaries, and lymphatic segments. The authors also performed a fractal analysis which revealed that this particular vascular network has an optimal spatial distribution and homogeneity occupying the entire fascia (fractal dimension = 1.063 ± 0.10; lacunarity = 0.60 ± 0.10).

Muscle vascularization has the key characteristic that it can have an enormous increase in blood flow during exercise to meet the 20- to 50-fold increase in demand for oxygen and substrates required to support the metabolic activities of active muscles. The low rate of blood flow per unit mass of tissue in quiescent skeletal muscles is due to relatively high basal vascular tone of primary arteries and arterioles that results from inherent myogenic tone as well as high activity of sympathetic nerves innervating them. The vascular tone depends on the level of free Ca_2+_ in the cytosol of vascular smooth muscle (VSM) cells and on physical forces imparted by distension secondary to changes in transmural pressure and by the flowing stream of blood [62]. The Ca^2+^ increases via release from the sarcoplasmic reticulum or influx into VSM cells from the extracellular compartment, result in the formation of the Ca^2+^–calmodulin complex. In response to these chemical and mechanical stimuli, endothelial cells release vasoconstrictor endothelin and the vasodilators prostacyclin (PGI2), nitric oxide (NO), carbon monoxide (CO), hydrogen sulfide (H_2_S), and endothelium-derived hyperpolarizing factor (EDHF), which initiate responses in the underlying smooth muscle. There is some evidence as to how the characteristics of the ECM can strongly influence the perfusion of the muscle. Hocking et al., have reported the mechanism wherein fibronectin fibrils, in connective tissue matrices, transduce signals from contracting skeletal muscle to local blood vessels to increase blood flow [63]. As a consequence, during exercise, local mechanisms in tissues cause arterioles to rapidly dilate to increase blood flow to tissues to meet the metabolic demands of contracting muscle. This mechanism, which is managed by connective tissue, permits a rapid and precise regulation of intramuscular blood flow. Similar results were found by Sarelius et al., (2016) who found how local arteriolar dilatation, produced by the contraction of skeletal muscle, is dependent upon extracellular matrix protein fibronectin-mediated vasodilatation [64]. Yamashiro & Yanagisawa (2020) have found how normal mechanical cues or altered responses to mechanical stimuli in endothelial cells and smooth muscle cells serve as the molecular basis of vascular diseases [65], supporting the concept that extracellular matrix maintains the structural integrity of the vessel wall and coordinates with a dynamic mechanical environment to provide cues to initiate intracellular signaling pathway. To prevent vascular disease development and progression, it is important to understand how vascular wall endothelial cells, smooth muscle cells, and fibroblasts sense and transduce hemodynamic force stimuli, such as shear stress, cyclic strain, and hydrostatic pressure, into intracellular biochemical signals. Shi and Tarbell (2011) found that both endothelial cells and smooth muscle cells utilize integrins to bind to their ECM [66]. Furthermore, besides fluid flow, mechanical stretch and pressure also affect smooth muscle cells and fibroblasts. Langevin et al., have also proposed that fibroblasts play a role in regulating extracellular fluid flow into the tissue and protect against swelling when the matrix is stretched. It was shown that, in response to static stretching of the tissue, fibroblasts expand within minutes by actively remodeling their cytoskeleton, contributing to the drop in tissue tension that occurs during viscoelastic relaxation [67].

Recently, RAS (renin–angiotensin system) components were found in the muscular fascia [68]. RAS is a peptidergic system with endocrine features that plays a key role in regulating blood pressure, but also in the progression of fibrosis [10,16]. According to Pirri et al., (2022) [68], Ang II type 1 (AT1R) seems to be the most expressed angiotensin receptor subtype (300.2 ± 317 copies/25 ng of mRNA), followed by MAS receptor (37.1 ± 39.56 copies/25 ng of mRNA), and type 2 receptor (AT2R) (147 ± 122 copies/25 ng of mRNA), whilst angiotensinogen, angiotensin-converting enzyme 1 (ACE1) and ACE2 were hardly detectable. These data are of much interest because the presence of angiotensin receptors in the fascial tissue, mainly AT1R, suggests a possible involvement of the fasciae in blood flow regulation. Moreover, ACE2 degrades Ang II to Ang 1–7, thus reducing Ang II effects in vasoconstriction, fibrosis, and sodium retention. Finally, this system can also stimulate remodeling and fibrogenesis of the ECM, and this can alter the physical coupling between the ECM and capillaries, as described before.

The last aspect that could underlie speaking about the relationship between fasciae and vessels is the strong connection between superficial veins and fasciae, as demonstrated by Caggiati et al. [69]. Around the great and small saphenous vein there is a fascia that, thanks to various fibrous septa, support from outside the venous wall and maintain the patency of the vessels. Therefore, to have a good venous return it is important that the fascia is in a healthy condition to permit the vein to stay open.

## 5. Embryogenic Development

All components of the movement system have a common embryonic origin. Connective tissue, muscles, bones, and muscular fasciae derive from the paraxial mesoderm, more precisely from the myotome, which is one of the structures that make up the somite [70]. The myotome gives rise to the muscle, deep fascia, and intramuscular fasciae (epimysium, perimysium, and endomysium) that are intimately linked in the embryo and the adult, suggesting that interactions between these tissues are crucial for their development [71]. Evidence proposes that it is the associated connective tissue cells of the muscle that pattern the myoblast to form the appropriate muscle [72] and connective tissue fibroblasts critically regulate two aspects of myogenesis: muscle fiber type development and maturation [71]. Tcf4+ fibroblasts have been shown to play an active role in chick and mouse limb muscle morphogenesis by establishing a prepattern structure of muscles, to which muscle precursor cells migrate and differentiate [72]. During myogenesis, high levels of tcf4+ expressing fibroblasts are found within the connective tissue of the myofascial unit [71]. Thus, it is the fascial tissue supporting the skeletal myocytes that provide the necessary pattern to allow development of the muscle [73]. Furthermore, innervation comes from the migration of neural crest cells to their somites, and in its turn is organized in myotomes. The development of the neural system is a major influencer, and at the same time, it is influenced by the fascial system, establishing an intimate relationship [74]. Thus, this embryogenic nervous relationship might be important for the synchronization of motion components.

## 6. Discussion and Clinical Applications

All of this new knowledge about intramuscular connective tissue and fasciae and their relation to muscle supports the concept of myofascial unit. Indeed, from this review it is evident that to generate efficient contraction, it is fundamental to have a harmonic activation of the motor units, a good vascularization of the muscle, and a good proprioception. All of these elements are linked to each other by the intramuscular connective tissue and the overlying fascia, which consequently could be considered like a bridge, or the glue, that connects all the elements involved in a movement (Figure 2). It also becomes evident that any alteration that affects one of these elements could damage the others as well, worsening the problems [75,76]. Only by understanding this strong interconnection can we really understand myofascial pain and its characteristics, and consequently propose a physically focused treatment (Figure 3). Indeed, its diagnosis and treatment are still debated. Recently, Fitzcharles et al., (2022) [77] wrote, “the concept that a regional musculoskeletal pain may occur in the absence of identifiable tissue abnormality may be puzzling...”, but in our opinion an answer can be found if we consider the fascia, and how it could itself be a cause of pain, or affect muscular contraction, the vascularization of the district, or the proprioception.

Therefore, we can consider myofascial pain as the clinical outcome of myofascial unit alteration. This new point of view requires a different management of patients: if we consider the intrinsic ability of the fascia to communicate and to spread tensions far away, it is evident that poor management of myofascial pain can cause a worsening of disability and the course of regional versus widespread musculoskeletal pain.

## 7. Conclusions

Among all of the elements involved in muscular contraction, intramuscular connective tissue is acquiring importance, and at the same time, motor unit can no longer be considered the only element for motion organization. This narrative review supports the hypothesis that all motions are generated by different important elements such as fascia (connective tissue), nerves (MS), and vessels (lymphatic and vascular). All of these elements work in synergy to better organize movements in the most efficient manner (Figure 1). The ability to modulate synergic motor units, localized in neighboring muscles, will facilitate movements, improve efficiency, and decrease joint stress.

In conclusion, to assess a more comprehensive model of movement organization, it seems reasonable to include the bilateral dependent relationship, both anatomical and functional, that occurs between fascia, muscle, and accessory elements, all included in the entity called the myofascial unit that represents the physiological basic element for peripheral motor coordination.

## Figures and Tables

**Figure 1 ijms-24-04527-f001:**
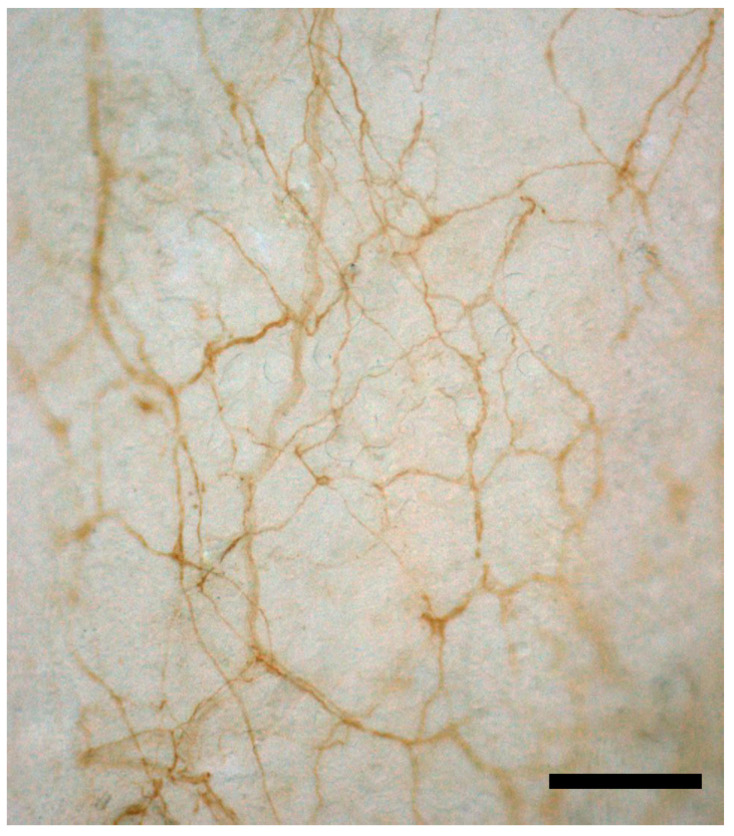
S100 immunohistochemistry reaction of thoracolumbar fascia of the mouse. The reaction shows a dense, thin neural network strongly connected with the extracellular matrix of the fascial tissue and consequently particularly responsive to stretch and tensional variations, highlighting the importance of fascia as a sensory system. Scale bar: 100 µm.

**Figure 2 ijms-24-04527-f002:**
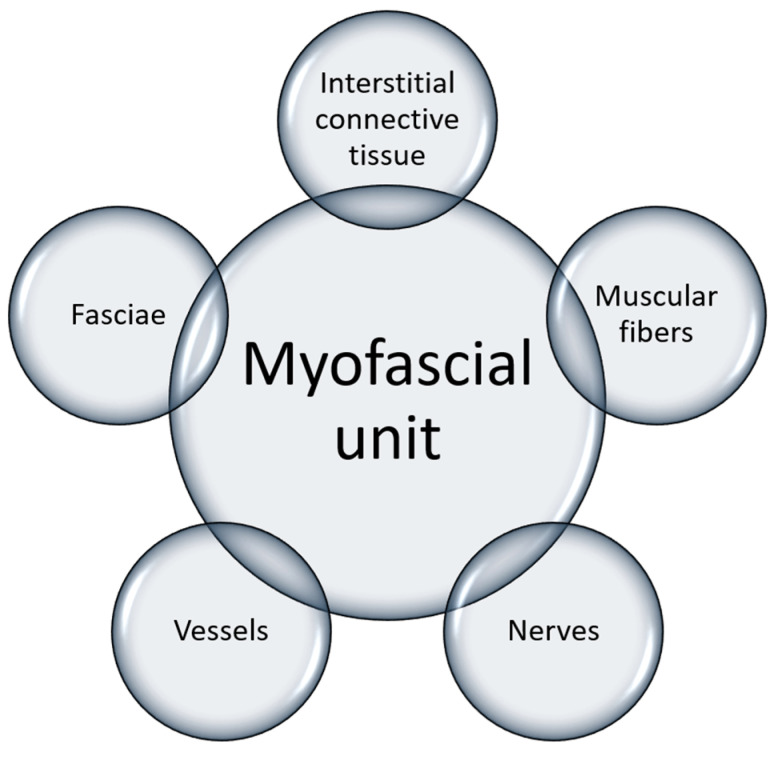
The myofascial unit composition.

**Figure 3 ijms-24-04527-f003:**
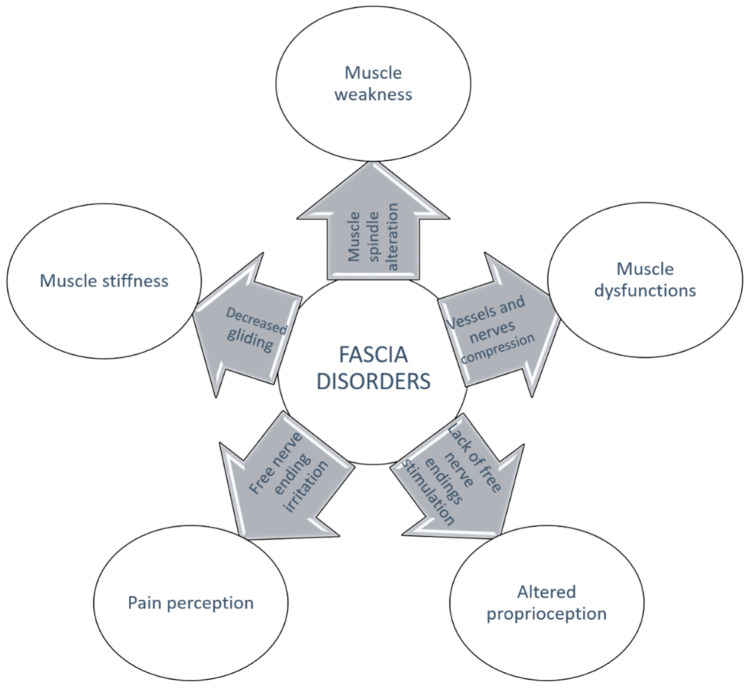
Any alteration affecting one of the interconnected elements of the MFU is at the basis of fascia disorders.

## Data Availability

Not applicable.

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
