# Peer review of "From Muscle to the Myofascial Unit: Current Evidence and Future Perspectives"

_ijms, 2023, doi:10.3390/ijms24054527_

Round 1

Reviewer 1 Report

With pleasure I read the present manuscript, dealing with a very appropriate holistic vision of different tissue and structure (namely, interstitial connective tissue, muscular fibers, fasciae, vessels and nerves) involvement in the movement control. I found the manuscript very interesting in this field, also from the instructional human anatomy point of view. The manuscript is written well, it is complete and the references are adequate. 

I just have some minor revision request:

Line 35: I suggest to replace the term “joints proprioceptive nerves” with the more appropriate “joint proprioceptors”.

Line 93: change take with taking

Line 93-94: I would suggest to replace "endomysium and epimysium" with a more generic "intramuscular connective tissue".

Line 181: the abbreviations (MS) should be explained at their first comparison throughout the text.

Author Response

We thank the Reviewer for the comments and the words spent on our work.

Authors followed the indications and modified as suggested (Lines 35-93-94), including also the explanation of abbreviation for muscles spindle (MS) and related abbreviations throughout the text. 

Reviewer 2 Report

I read with great pleasure the paper by Stecco et al. which reviewed the scientific literature in support of the use of the new term “myofascial unit” as a fundamental element in peripheral motor control. Overall, the paper is well written, fluent and deals with a hot topic in the field of musculoskeletal system, both in physiological and pathological conditions. I would have only one minor suggestion. Since the Authors empathized how the fascia may be viewed as a sensory system due to the dense sensory nociceptive innervation and thus playing a key role in pain perception and in pathogenesis of several musculoskeletal dysfunctions, in the reviewer’s opinion, the addition of some figures regarding this key element of the fascial system would add significance to the paper (see for instance Ryskalin et al. Life (Basel). 2022 May 17;12(5):743).

Author Response

Authors thank the Reviewer and added a Figure showing the dense neural fascial network as suggested (Figure 1 in the revised text).